# Distinctive phases and variability of vibration-induced postural reactions highlighted by center of pressure analysis

**Mohamed Abdelhafid Kadri**[1], **Emilie Bouchard**[1], **Lydiane Lauzier**[1], **Hakim Mecheri**[2], **William Bégin**[1], **Martin Lavallière**[1], **Hugo Massé-Alarie**[3], **Rubens A. da Silva**[1,4], **Louis-David Beaulieu**[1]*

**1** Lab BioNR, Centre intersectoriel en santé durable, Université du Québec à Chicoutimi, Chicoutimi, QC, Canada, **2** Institut de recherche Robert-Sauvé en santé et en sécurité de travail, Montréal, QC, Canada, **3** Centre Interdisciplinaire de Recherche en Réadaptation et Intégration Sociale, Université Laval, Québec, QC, Canada, **4** Services gériatriques spécialisés–Hôpital de la Baie, Centre Intégré de Santé et Services Sociaux du Saguenay—Lac-Saint-Jean, Saguenay, QC, Canada

* louis-david_beaulieu@uqac.ca

**Data Availability Statement:** Anonymous minimal data set is available at the following institutional repository (upon request): https://doi.org/10.5683/SP3/ZICB86. Ethical restrictions apply since

## Abstract

### Background

The vibration-induced postural reaction paradigm (VIB-PR) offers a unique way for investigating sensorimotor control mechanisms. Measures of VIB-PR are usually calculated from the whole VIB period, yet recent evidence proposed that distinctive mechanisms are likely at play between the early vs. later phases of the postural reaction.

### Objectives

The present work verified if spatiotemporal analyses of center of pressure (COP) displacements can detect differences between these early/later phases of VIB-PR. Also, we further characterized the intra/inter-individual variability of COP measurements, since the underlying variability of VIB-PR remains largely unexplored.

### Methods

Twenty young volunteers realized two experimental conditions of bipodal stance with eyes closed: (i) bilateral VIB of tibialis anterior (TIB) and (ii) Achilles' (ACH) tendons. Each condition consisted of five trials and lasted 30 s as follows: 10 s baseline, 10 s VIB and 10 s post-VIB. Linear COP variables (antero-posterior (AP) amplitude & velocity) were computed for both VIB and post-VIB periods using the following time-windows: early 2 s, the later 8 s and the whole 10 s duration. Intra- and inter-individual variability were respectively estimated using the standard error of the measurement and the coefficient of variation. Both variability metrics were obtained using five vs. the first three trials.

### Results

Significant contrasts were found between time-windows for both VIB and post-VIB periods. COP variables were generally higher during the early 2 s phase compared to the later 8 s

**Funding:** MAK's postdoctoral fellowship was supported by the Centre intersectoriel en santé durable (no grant number - http://www.uqac.ca/santedurable/). LDB research program is supported by the Fonds de Recherche du Québec – Santé (Research Scholar Junior 1; #297854, https://frq.gouv.qc.ca). The funders had no role in study design, data collection and analysis, decision to publish, or preparation of the manuscript.

**Competing interests:** The authors have declared that no competing interests exist.

phase for both TIB [mean difference between 8 s– 2 s phases: Amplitude AP = -1.11 ± 1.14 cm during VIB and -2.99 ± 1.31 during post-VIB; Velocity AP = -1.17 ± 0.86 cm/s during VIB and -3.13 ± 1.31 cm/s during post-VIB] and ACH tendons [Amplitude AP = -0.37 ± 0.98 cm during VIB and -3.41 ± 1.20 during post-VIB; Velocity AP = -0.31 ± 0.59 cm/s during VIB and -3.89 ± 1.52 cm/s during post-VIB]. Most within- and between-subject variability scores were below 30% and using three instead of five trials had no impact on variability. VIB-PR patterns were quite similar within a same person, but variable behaviors were observed between individuals during the later phase.

## Conclusion

Our study highlights the relevance of identifying and separately analyzing distinct phases within VIB-PR patterns, as well as characterizing how these patterns vary at the individual level.

## Introduction

Vibration of tendons is an effective method to non-invasively depolarize muscle spindles. In the absence of vision, vibration induces postural reactions (VIB-PR) when applied to muscles involved in the ongoing postural task [1, 2]. For instance, vibration of the Achilles tendons when standing upright sends the same sensory information that would have occurred if the ankles' plantar flexors were stretched. This false proprioceptive signal is interpreted as a forward fall and a backward postural reaction occurs to maintain balance. The VIB-PR approach was used for the past decade to investigate mechanisms involved in sensorimotor processing and postural control [3, 4], and how these mechanisms can be altered by ageing [5, 6], neurological [7] or musculoskeletal conditions [8].

VIB-PR are often measured on a force platform using variables related to center of pressure (COP) displacements. Linear COP variables measured during different postural conditions without vibration are recognized as valid and reliable to assess postural control and to provide an estimation for the risk of falling [9–11]. For example, higher amplitude or velocity of COP displacements observed in older individuals during a bipodal stance would likely reflect less efficient postural control mechanisms and increased risk of falling [9]. However, the interpretation of COP-related variables in the context of VIB-PR is less evident. VIB-PR of lower amplitude or slower velocity could suggest reduced sensitivity of muscle spindles and/or an altered processing of sensory signals by neural networks involved in postural control [12, 13]. Alternatively, lower postural reactions could also result from a top-down inhibition of sensory gains meaning that the postural control system was more efficient to regain control over the imbalance caused by the disruptive sensory information generated by vibration [14]. These different mechanisms are most likely impossible to disentangle if COP variables are analyzed and interpreted the same way they are for static postural tasks, which calls for different ways of measuring VIB-PR.

We recently reported distinct phases within VIB-PR trajectories [2]. Indeed, VIB-PR comprised an early phase, occurring within the first 2 s after vibration start, and consisting of a rapid and stereotyped COP displacement, followed by an 8 s slowing phase during which the COP position tended to stabilize [2]. Such dynamic patterns remained undetected by COP variables calculated for the whole 10 s vibration period which is usually the way COP

displacements are analyzed. Measuring and comparing distinct phases of COP trajectories seem promising for unravelling the different mechanisms underlying VIB-PR phenomenon [2, 15]. Furthermore, VIB-PR traces seemed to vary between individuals, especially during the later re-stabilization phase [2]. The underlying variability of VIB-PR still remains largely unexplored and could greatly affect the reliability, and hence the validity, of any attempt to measure VIB-PR. This issue is mostly overlooked when choosing VIB-PR methods and analyses (e.g., how many trials to account for variability, most reliable COP measure, etc.). Further research in this field is critically needed to gain a better insight on how VIB-PR traces vary within and between-individuals and adjust accordingly the methodological procedures to account for this variability in COP-related measurements.

The primary objective of the present study was to compare spatiotemporal variables of COP displacements caused by bilateral vibration of ankle tendons (tibialis anterior-TIB and Achilles-ACH) using three different time-windows within VIB-PR, i.e. the early 2 s after VIB start (early imbalance phase), the later 8 s (re-stabilization phase) and the whole 10 s vibration period (usual measurement method). Our main hypotheses are: (i) the early phase of rapid imbalance in the first 2 s after VIB start will be characterized by higher COP amplitude and velocity compared to the later 8 s phase of COP re-stabilization and (ii) lower COP displacement and velocity will be found for the later 8 s phase compared to the whole 10 s period. In order terms, a separate analysis of these two distinct phases within VIB-PR will result in more contrasting COP data compared to a global analysis of the whole vibration period. A secondary objective was to characterize the variability (within and between-subject) of these COP measurements and propose recommendations to account for the observed variability in future research using the VIB-PR approach. Considering our previous research [2], we hypothesized that variability of VIB-PR will be lower with longer measurement periods (i.e. whole 10 s vs. last 8 s vs. early 2 s). Indeed, reliability metrics in our previous work systematically improved when increasing time-windows of measurement. A descriptive analysis based on visual observations of VIB-PR traces will also be used to further meet the study's objectives.

## Material and methods

### Participants

Twenty healthy young volunteers (11 men, 9 women) were recruited. Participants had no prior history of neurological, musculoskeletal, vestibular disorders or ankle, knee and hip injuries in the past 2 years. Participants gave their written informed consent in accordance with the Declaration of Helsinki and the local research ethic committee (#2019–240). Participants' characteristics were (mean ± standard deviation): age = 24.8 ± 2.4 years; height = 171.4 ± 9.4 cm; body mass = 73.7 ± 16.0 kg; level of total physical activity (global physical activity questionnaire-GPAQ$_{v2}$ [16]) = 3742 ± 2971 MET-minutes/week (MET = Metabolic Equivalent of Task).

### Experimental procedure

All measurements were done in a controlled laboratory environment (Lab BioNR). The experimental session consisted of 2 conditions realized in a random order between participants: (i) bilateral vibration of tibialis anterior tendons (TIB) and (ii) bilateral vibration of Achilles tendons (ACH). Five trials per condition were done (with 1 min rest between trials and approximately 5 min rest between conditions while participants sat on a chair and vibrators were repositioned over the next targeted tendons). Participants were asked to stand barefoot with eyes blindfolded and the arms by their sides along the body on a force platform (BIOMEC400, EMG System do Brasil, Ltda., SP, Brazil). The feet position was standardized according to

precise marks on the platform. Precisely, participants were first asked to position their feet hip-width apart with their heels aligned on tape markers placed posteriorly along the X axis. Then, big toes were aligned on the Y axis and tape markers were added along the 5th metatarsal. Care was taken to ensure that this standardized position was kept throughout the experiment. Two vibrators (VB115, Techno Concept, France) were installed bilaterally with elastic straps on the targeted tendons. Tendons were identified by palpation and slight contraction (ACH tendon located proximally to the calcaneal tuberosity and TIB on the anteromedial surface of the ankle above the talus and near the medial malleolus). Each vibrator comes with a small depression at its center in which the tendon is placed for improving comfort and stability of the device. For both ACH and TIB conditions, vibrators were localised approximately at the level of medial and lateral malleoli. Vibration was activated at 80 Hz frequency and 1mm amplitude [4, 17]. There is no published guideline for the best duration of vibration to be used for eliciting postural reactions. We chose 10 s based on our previous study which was able to induce strong and reliable reactions and was long enough for most participants to reach the re-stabilisation phase [2]. Posturography measurements were recorded for a total duration of 30 s as follows: 10 s quiet standing before vibration, 10 s vibration of either TIB or ACH tendons (VIB) and a further 10 s after stopping vibration (Post-VIB). An investigator stood close to the participant to avoid falls during the experiment.

## Data processing and analysis

Reaction force signals from the force platform were acquired at 100 Hz and filtered with a 35 Hz second-order Butterworth filter and converted into COP data [2]. MATLAB scripts (The Mathworks, Natick, MA, USA) were used to calculate the selected linear COP (Amplitude and Velocity) variables.

**Descriptive analysis of VIB-PR traces.** The precise time course of COP displacements was first processed by re-centering COP position at [0;0] coordinates by subtracting COP data to the first measurement obtained at 0.01s. All further COP displacements in the antero-posterior (AP) were therefore expressed relative to this initial COP position. COP position was plotted across time to give a general appreciation of COP trajectories. A descriptive analysis of VIB-PR patterns and inter/intra-individual variability was realized based on mean group as well as all individual COP traces.

**Spatiotemporal analysis of COP data.** Stabilographic analysis of COP excursions focussed on y-axis data since bilateral vibration of ACH and TIB tendons result in antero-posterior displacements [2]. The chosen linear COP variables consisted of: (i) *Amplitude AP* (the absolute distance between the max and min COP displacement, in cm) and (ii) *Velocity AP* (sum of the cumulated COP displacement divided by the total time, in cm s−1) [18]. These COP variables were computed separately for VIB and Post-VIB periods and for the following time-windows: (1) *First 2 s* (early imbalance phase); (2) *Last 8 s* (later phase of re-stabilization); and (3) the *Whole 10 s* block–for more clarity please refer to Fig 1A for a visual illustration of these time-windows (identified as W1, W2 and W3 in the Figure, respectively).

## Statistical analysis

Statistical analysis was done using SPSS version 25 (Armonk, NY, USA) and GraphPad Prism version 7.00 (La Jolla California USA). Data normality was confirmed using the Shapiro-Wilk test and visual screening of box-and-whisker, normal Q-Q and detrended Q-Q plots. Of note, only one variable from ACH condition was identified as non-normal by the Shapiro-Wilk test (*Velocity AP*, p = 0.04) for the 0–10 sec time block after vibration start. Visual screening of plots revealed that a single participant explained the Shapiro-Wilk result. This participant had

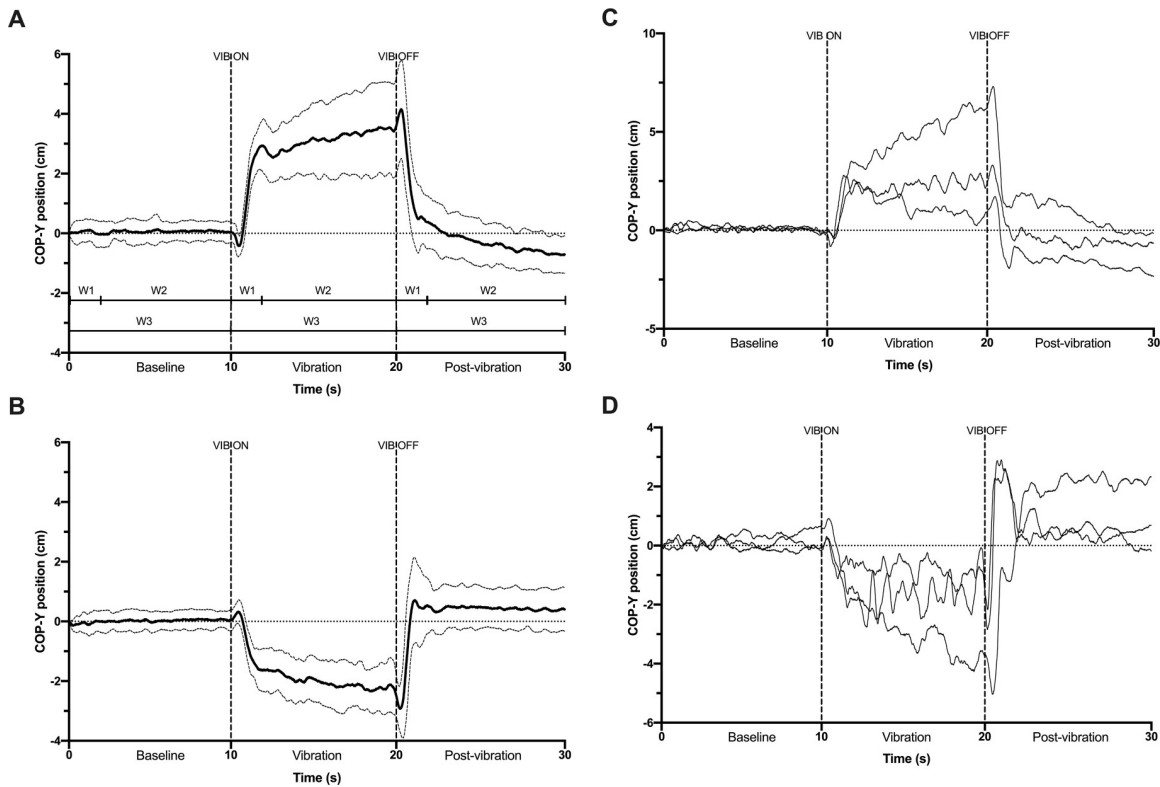

**Fig 1. Graphical representation of VIB-PR for TIB (A, C) and ACH (B, D) conditions.** Graphs A & B illustrate group mean and standard deviation of COP position in the anteroposterior axis (COP-Y position). Graphs C & D illustrate examples of mean individual VIB-PR traces (i.e. their mean trace from the five trials) from three participants presenting with distinctive patterns. VIB = vibration; W1 = the *First 2 s* time-window; W2 = the *Last 8 s* time-window; W3 = the *Whole 10 s* time-window.

data identified by SPSS as far out, but not outlier. Therefore, we decided not to remove the participant from the sample and to use parametric analysis since the remaining variables respected the statistical prerequisites.

**Primary objective–comparing COP data between different VIB-PR phases.** The mean of 5 trials was first calculated per individual for each variable. To evaluate if the selected COP variables differed between time-windows (*First 2 s* vs. *Last 8 s* vs. *Whole 10 s*), a repeated measure ANOVA was applied. When a significant within-subject effect was found, pair-wise comparisons (paired samples t-test) were realised. The level of statistical significance was set at $p < 0.05$ and effect sizes (Cohen's d ES) were calculated (G*Power 3.1 Software) whenever relevant to give a perspective about the magnitude of the effects (large if $>0.8$, moderate if $>0.5$, small if $>0.2$ and null if $<0.2$).

**Secondary objective–variability of VIB-PR.** Within-subject (or trial-to-trial) variability was investigated using the Standard Error of the Measurement (SEMeas, expressed in the same unit as the measure). This metric has the advantage of being completely independent of between-subject variance [19]. In other terms, it computes the level of error per individual across the five trials and then produces a standard estimation based on the whole sample. Also, comparisons of SEMeas between outcome measures with different units is possible when normalizing to the group mean (%SEMeas = (SEMeas/mean)*100). Between-subject (or Group) variability was evaluated using standard deviations (SD) and the Coefficient of variation (CV = (SD/mean)*100) which estimates the level of variation across individuals [20]. Metrics of individual variability (SEMeas, %SEMeas) and group variability (SD and CV) were calculated

separately for the three time-windows (*First 2 s*, *Last 8 s*, *Whole 10 s*) for VIB and Post-VIB periods, and were computed using data from the five trials. However, to further evaluate the global impact of the number of trials on variability, data from ACH and TIB conditions were pooled and variability metrics were calculated and compared using five versus the first three trials, since three trials are often used in VIB-PR studies.

## Results

### Descriptive analysis of VIB-PR traces

Fig 1A & 1B shows group mean and standard deviation for TIB & ACH conditions, respectively. As expected, the transition between the early and later phases occurred within the first 2 sec after VIB start for both TIB and ACH conditions. Standard deviations around the mean group trace remained small during the early phase, then increased significantly during the later phase, especially for TIB condition. Post-VIB period showed similar patterns of distinctive early/later phases but in the opposite direction compared to VIB. COP position did not return to baseline (BL) position even 10 s after VIB cessation, which was particularly evident for TIB condition. Furthermore, the early reaction for both VIB and Post-VIB periods started with a small COP displacement in the reverse direction observable during the first milliseconds after VIB start and cessation (Fig 1A & 1B). For example, Fig 1 A shows that when VIB is activated, COP position in fact goes posteriorly for about 0.5 s before quickly moving anteriorly in the following seconds.

Examples of individual traces for six participants presenting with different patterns of postural reaction are depicted in Fig 1C (TIB condition) & 1D (ACH condition). Participants generally showed similar COP displacements during the early phase after VIB start, but not during the later 8 s phase which was either characterized by a continuous displacement of COP position, a relative re-stabilization (even though quite variable) around a particular COP position or even a cancellation of the early VIB effect and progressive return toward baseline position. A careful examination of all VIB-PR traces from the whole sample (n = 200 traces, not presented here) showed that they all complied with either one of these three patterns. Also, as shown in Fig 2 (illustrating the five VIB-PR trials and their mean for the six participants), VIB-PR patterns were most often similar within a same individual. The mean trace tended to reduce inter-trial variation of COP position while representing the general underlying pattern.

### Differences between *First 2s*, *Last 8 s* and *Whole 10 s* time-windows

Table 1 presents mean ± SD of COP variables selected for the present study and Table 2 details results for statistical comparisons between time-windows for the tested conditions of vibration, including mean differences and Cohen's d effect sizes. ANOVA revealed significant effects (p < 0.05) for all comparisons between time-windows during VIB and Post-VIB periods. In general, pair-wise comparisons between the three time-windows reached the level of statistical significance and had moderate to high effect sizes, with similar findings observed for VIB and Post-VIB periods. *Amplitude AP* was highest during the *Whole 10 s* block, followed by *First 2 s* and *Last 8s* (except during VIB of ACH which was not different between *First 2 s* and *Last 8 s* time-windows). Conversely, *Velocity AP* had highest values during the *First 2 s*, followed by *Whole 10 s* and lowest values obtained for *Last 8 s.*

### Variability of VIB-PR

**Within-subject variability.** SEMeas and %SEMeas are detailed in Fig 3. The *First 2 s* time-window had the highest SEMeas for both COP variables, especially during Post-VIB

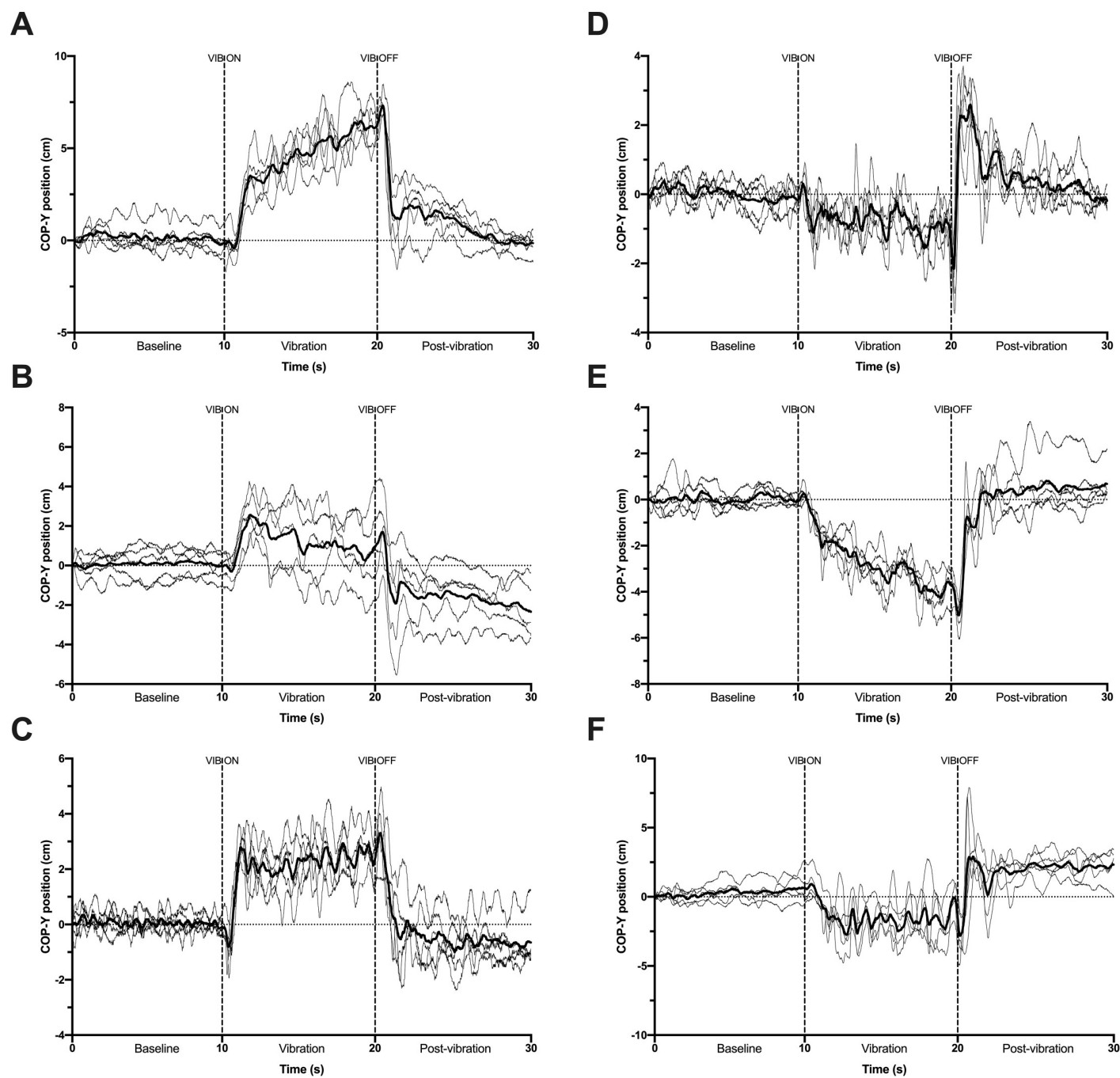

**Fig 2. Graphical representation of individual VIB-PR traces for TIB (A-C) and ACH (D-F) conditions.** Each graph represents a different participant (the same selected for Fig 1C & 1D), light grey lines represent the five VIB-PR trials and the bold line represents the mean trace calculated from these five trials. VIB = vibration.

period. The lowest SEMeas (Fig 3A & 3C) were found for the *Last 8 s* time-window, except during VIB of ACH which had similar error scores than the *Whole 10 s*. When these variability metrics were expressed relative to their respective pooled mean (Fig 3B & 3D), contrasts between time-windows became less apparent. The *First 2 s* mostly remained higher except for *Amplitude AP* during Post-VIB period of both TIB and ACH conditions which was slightly higher for the *Last 8 s*. %SEMeas remained mostly between 15–30% of variation (ranging

**Table 1. Group results for the selected variables, conditions and time-windows analyzed.**

| Conditions | Tibialis anterior | | Achilles | |
|---|---|---|---|---|
| Periods | Vibration | Post-vibration | Vibration | Post-vibration |
| | mean ± SD | mean ± SD | mean ± SD | mean ± SD |
| **Amplitude AP (cm)** | | | | |
| First 2 s | 4.43 ± 0.90 | 5.24 ± 1.34 | 2.97 ± 0.70 | 5.67 ± 1.32 |
| Last 8 s | 3.33 ± 0.97 | 2.25 ± 0.49 | 3.33 ± 0.72 | 2.26 ± 0.37 |
| Whole 10 s | 5.77 ± 1.48 | 6.13 ± 1.45 | 4.50 ± 0.67 | 5.94 ± 1.26 |
| **Velocity AP (cm/s)** | | | | |
| First 2 s | 3.83 ± 0.82 | 4.73 ± 1.42 | 2.84 ± 0.72 | 5.48 ± 1.58 |
| Last 8 s | 2.66 ± 0.79 | 1.60 ± 0.41 | 2.54 ± 0.72 | 1.59 ± 0.35 |
| Whole 10 s | 2.90 ± 0.72 | 2.23 ± 0.52 | 2.60 ± 0.68 | 2.37 ± 0.48 |

SD = standard deviation; AP = anteroposterior axis; s = seconds

14.60–31.50) and were similar between COP variables. Finally, reducing the number of VIB-PR trials from five to three had no significant impact on measurement error (mean 1.05 times higher SEMeas when using three instead of five trials, ranging 0.97–1.17).

**Group variability.** SD and CV are presented in Fig 4. Group standard deviations (Fig 4A & 4C) were quite similar across the three time-windows during both VIB conditions, except for *Amplitude AP* during VIB of TIB which was about 1.5 times higher for the *Whole 10 s*. Conversely, more contrasts were found between time-windows during post-VIB periods. For

**Table 2. Results of statistical comparisons between time-windows for the tested conditions of vibration.**

| | Tibialis anterior | | Achilles | |
|---|---|---|---|---|
| | Vibration | Post-vibration | Vibration | Post-vibration |
| | mean difference ± SD | mean difference ± SD | mean difference ± SD | mean difference ± SD |
| | ES (p-value) | ES (p-value) | ES (p-value) | ES (p-value) |
| **Amplitude AP (cm)** | | | | |
| $ANOVA_{RM}$ | | | | |
| F(df; df error); p-value | 53.7(2; 38); <0.001 | 136.8(1.2; 23.4); <0.001 | 42.3(1.5; 28.5); <0.001 | 178.0(1.1; 21.1); <0.001 |
| Whole 10 s–First 2 s | 1.34 ± 0.97 | 0.89 ± 0.51 | 1.53 ± 0.66 | 0.27 ± 0.32 |
| | 1.37 (<0.001) | 1.76 (<0.001) | 2.32 (<0.001) | 0.84 (0.001) |
| Whole 10 s–Last 8 s | 2.44 ± 1.05 | 3.88 ± 1.29 | 1.17 ± 0.65 | 3.68 ± 1.14 |
| | 2.33 (<0.001) | 3.02 (<0.001) | 1.79 (<0.001) | 3.22 (<0.001) |
| Last 8s –First 2 s | -1.11 ± 1.14 | -2.99 ± 1.31 | 0.37 ± 0.98 | -3.41 ± 1.20 |
| | 0.97 (<0.001) | 2.28 (<0.001) | 0.37 (0.110) | 2.85 (<0.001) |
| **Velocity AP (cm/s)** | | | | |
| $ANOVA_{RM}$ F(df; df error); p-value | 37.1(1; 19); <0.001 | 113.9(1; 19); <0.001 | 5.4(1; 19);0.031 | 132.0(1; 19) <0.001 |
| Whole 10 s–First 2 s | -0.94 ± 0.69 | -2.50 ± 1.05 | -0.24 ± 0.47 | -3.11 ± 1.21 |
| | 1.36 (<0.001) | 2.39 (<0.001) | 0.52 (0.032) | 2.57 (<0.001) |
| Whole 10 s–Last 8 s | 0.24 ± 0.17 | 0.63 ± 0.26 | 0.06 ± 0.12 | 0.78 ± 0.30 |
| | 1.38 (<0.001) | 2.39 (<0.001) | 0.54 (0.025) | 2.57 (<0.001) |
| Last 8s –First 2 s | -1.17 ± 0.86 | -3.13 ± 1.31 | -0.31 ± 0.59 | -3.89 ± 1.52 |
| | 1.36 (<0.001) | 2.39 (<0.001) | 0.52 (0.031) | 2.57 (<0.001) |

SD: standard deviation; ES: effect size (Cohen's d); AP: anteroposterior; $ANOVA_{RM}$: repeated measures analysis of variance; df: degrees of freedom

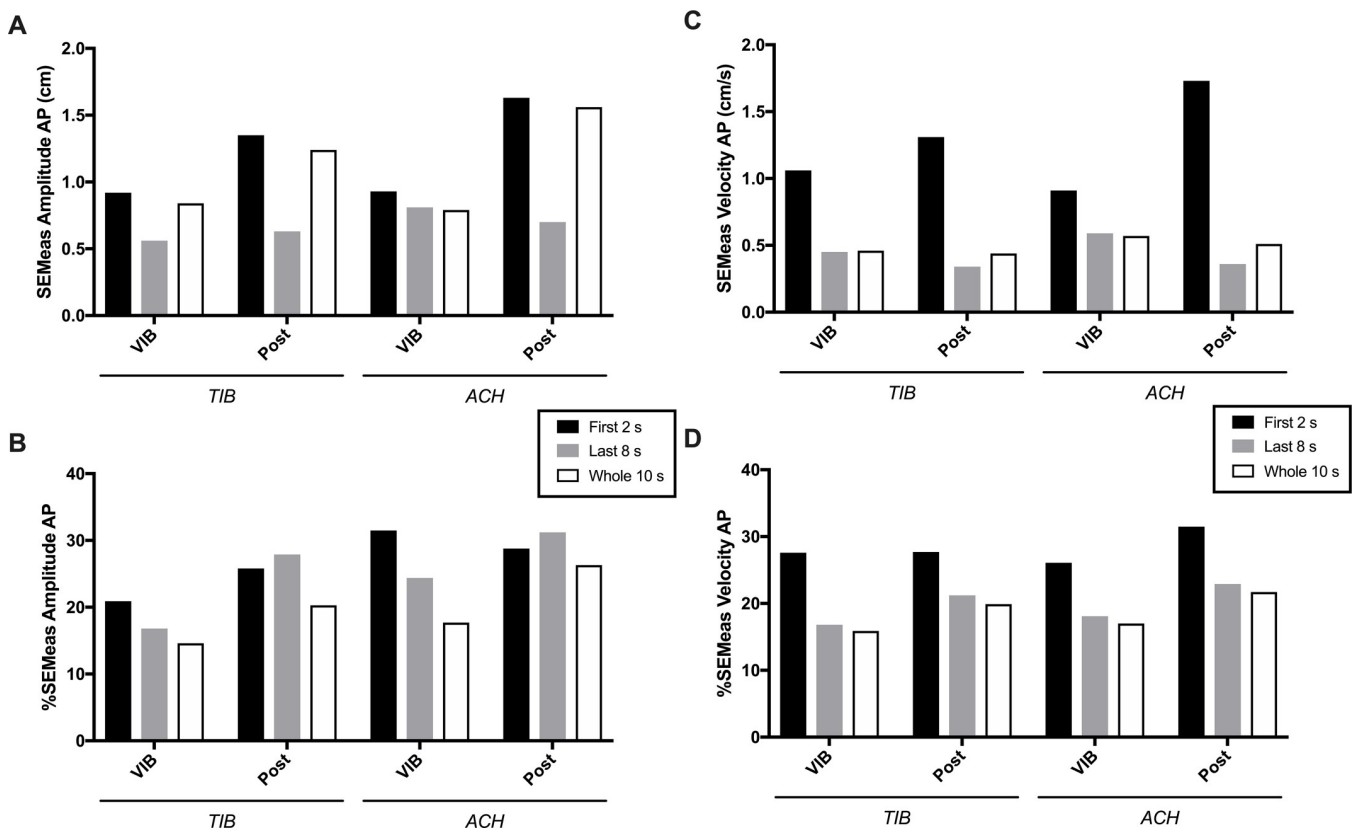

**Fig 3. Intra-individual variability of Amplitude (A & B) and Velocity (C & D) center of pressure variables for the three time-windows analyzed.** Variability is estimated by the standard error of the measurement expressed in the related unit of the measure (SEMeas) and in percentage of the pooled mean (%SEMeas) for. AP = anteroposterior axis; VIB & POST = vibration and post-vibration periods; TIB & ACH = tibialis anterior & Achilles' conditions.

*Amplitude AP*, *First 2 s* and *Whole 10 s* windows were about 3.0 times higher than *Last 8 s* for both TIB and ACH conditions. For *Velocity AP*, only the *First 2 s* remained higher, with scores 2.7–4.5 times higher than *Last 8 s* and *Whole 10 s* time-blocks. When comparing VIB and post-VIB periods, group SD tended to increase at post-VIB for *First 2 s* of TIB and ACH conditions (ranging 1.5–2.2 times higher). Conversely, a 2-fold decrease (ranging 1.9–2.1) of group SD was found for *Last 8 s* during post-VIB vs. VIB periods. No clear pattern was found for the *Whole 10 s* which either remained similar, increased or decreased at post-VIB compared to VIB periods. The same occurred for *Amplitude AP* of the *Whole 10 s*. Coefficient of variations (SD expressed relative to the group mean, Fig 4B & 4D) resulted in group variability ranging 14.81–30.08% with no clear contrast between COP measures or time-windows. Also, reducing the number of VIB-PR trials from five to three had no significant impact on between-subject variability (mean 1.02 times higher when using three instead of five trials, ranging 0.99–1.08).

## Discussion

Results from the present study confirmed our primary hypotheses since spatiotemporal analyses of COP displacements mostly revealed stronger postural reactions during the early 2 s phase, whereas the later 8 s phase was characterized, in general, by a progressive slowing of COP displacements. Most within- and between-subject variability scores for the selected COP variables were below 30% and using three instead of five trials had no impact on variability, hence advancing our understanding of the proposed approach from methodological and

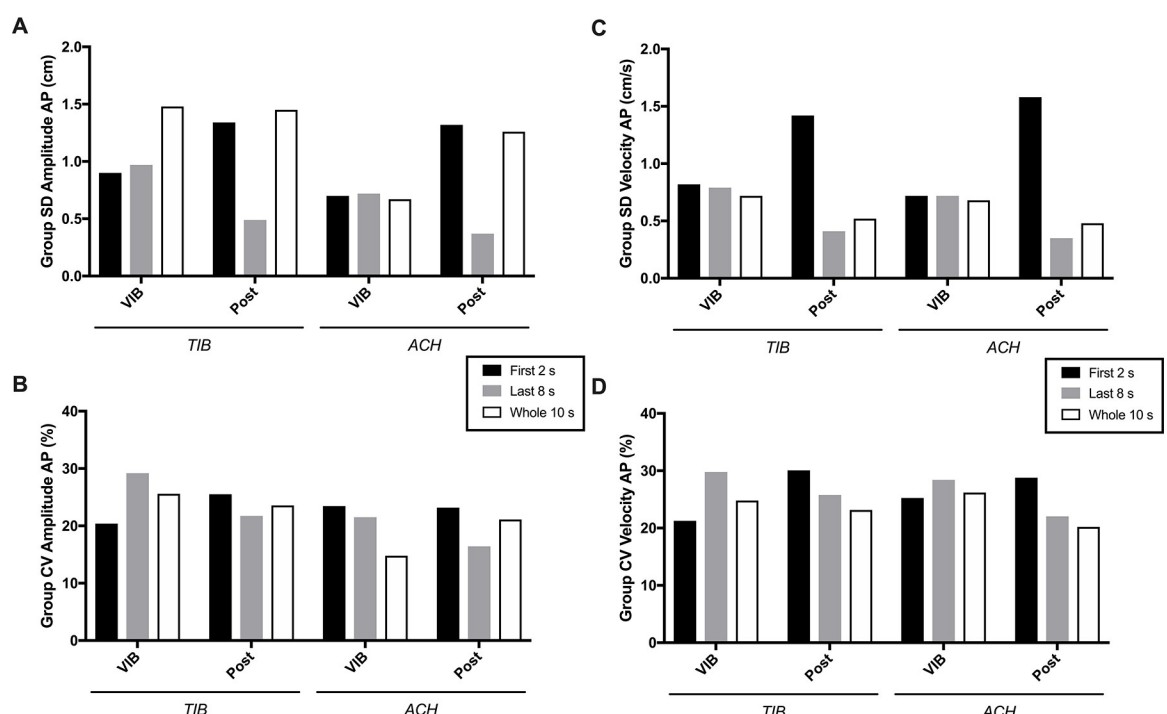

**Fig 4. Inter-individual (group) variability of Amplitude (A & B) and Velocity (C & D) center of pressure variables for the three time-windows analyzed.** Variability is estimated by the standard deviation (SD) and coefficient of variation (CV = SD in percentage of the mean). AP = anteroposterior axis; VIB & POST = vibration and post-vibration periods; TIB & ACH = tibialis anterior & Achilles' conditions.

metrological standpoints. VIB-PR patterns for the early and later phases were quite similar within a same person (Fig 2), but distinctive behaviors were observed between individuals during the later phase, suggesting that this VIB-PR phase could be more sensitive to discriminate individuals presenting with different patterns of reactions to a sensory disturbance. Overall, our study highlights the relevance of identifying and separately analyzing distinct phases within VIB-PR patterns, as well as characterizing these patterns at the individual level.

## Distinct phases within VIB-PR

Duclos et al. [15] characterized the time course of COP displacements induced by bilateral vibration of Achilles' tendon for 20 s at 80 Hz. They found that the COP position shifted posteriorly for approximately the first 16 s and then stabilized for the last 4 s. Maximal AP amplitude was observed between 2.35–4.8 s after VIB started and velocity increased significantly during the first 8 s and then remained similar for the rest of the vibration duration. Their results thus also highlight a dynamic evolution of COP shifts with a first phase of imbalance followed by a progressive re-stabilization. Compared to our study however, the contrast between the early and later phases is less apparent, the re-stabilization phase occurs later on and the velocity of COP displacements does not decrease during the later phase like we observed. These discrepancies may be explained by methodological differences: (i) they analyzed and compared COP variables using time-blocks of 4 s, hence merging the early 2 s phase with part of the later phase; (ii) they illustrated COP shifts also using 4 s time-points (i.e. 1 COP coordinate per 4 s) instead of the raw COP signal at 0.01 s precision used in the present work; (iii) they applied vibration at lower amplitude (0.2–0.5 mm) compared to ours (1 mm) which influences the

strength of the stimulus and of the resulting proprioceptive afferents [21]. Another study applied 1 mm vibration of the Achilles' tendon at 60 Hz for 10, 20 and 30 s and illustrated the resulting raw COP excursions (time precision/sample frequency not specified) like we did [22]. Similar patterns of early/later phases during VIB than those found in our study can be observed in their Fig 1 (single participant) and 2 (average group data). They rather coined the early period as a 'transition' phase and noted that it occurred during the first 2 s after VIB started but removed this phase from their global analyses of COP amplitude and velocity. Kavounoudias et al. [1] vibrated the tibialis anterior tendons during 3 s at different frequencies, including 80 Hz, and 0.2–0.5 mm amplitude [1]. Although 3 s was not long enough to reach re-stabilization as showed in the present work and others [2, 15], it is possible to observe in their Fig 1 (group data, time-precision/sample frequency not specified) that COP displacements began to slow down near 2 s after VIB start. Altogether, these results from the literature confirm the reproducibility of our findings about the early/later patterns observed in healthy adults. It also stresses out the critical impact of VIB parameters (e.g. amplitude and duration) and time-precision of COP analyses for appropriately detecting and measuring distinctive VIB-PR behaviors. It also suggests that the duration of the early and later phases depends on the selected parameters and a visual inspection of the COP excursion may be critical to determine the optimal duration of these phases.

Different underlying mechanisms are likely at play between the early and later patterns of VIB-PR. Based on the inverted pendulum [11] and sensory re-weighting theories [14, 23] for postural control, the sensory information originating from VIB of ankle tendons first results in a quick response from the vibrated muscle to counteract their perceived stretching and maintain balance. This first response was evident in Fig 1 (and in others [1]), that is, the small COP displacement in the opposite direction during the first 0.5 s after VIB start. For example, when vibrating TIB tendons a normal postural control system would interpret this unexpected disturbance as a backward fall (i.e. TIB are stretching while the knee, hip, and trunk joints remain still). In response, the contraction of TIB muscles causes a backward rolling of the feet and higher pressure applied under the heels. Previous work using electromyographic recordings concluded that this contraction in fact happens at relatively long latencies and would therefore involve distributed control networks within the central nervous system instead of only a sum of spinal reflexes [1, 3]. This short contraction is then strong enough to accelerate the center of mass forward, hence causing an actual forward fall along with compelling sensory evidence of this fall from different sources (foot soles, vestibular system, ankle plantar flexors, etc.). To avoid reaching the limits of stability, the postural control system would have to upregulate the processing of the most reliable sources and down-regulate the disturbing ones. This phenomenon is called sensory-reweighting and involves complex neural processes that are still incompletely understood [14, 23]. The important COP displacement observed in the early 2 s would likely represent this first attempt of down-regulating sensory information from the vibrated muscles. In our example of TIB vibration, the COP moves quickly from slightly posterior to its baseline position 0.5 s after VIB start to a more anterior position toward toes and forefoots as the plantar flexors contract to decelerate the forward movement of the center of mass. The fact that we observed low standard deviations around the group mean for this part of the VIB-PR (Fig 1) suggest that it likely engaged automatic postural responses to react rapidly to the unexpected 'imbalance'. Based on the literature, sensory information originating from the vestibular system, skin mechanoreceptor from foot soles or muscle spindles from non-vibrated muscles were likely involved in this dynamic re-weighting of afferents to maintain postural control [1, 15, 24].

Mechanisms involved afterwards during the later phase are less evident to explain without additional neurophysiological outcome measures. Under continuous VIB, the postural control

system would have to find a new equilibrated state through this contradicting multisensory information. Reaching a stabilized COP position during vibration would require complex sensorimotor control mechanisms leading to plastic (synaptic and/or morphologic) adaptations of spinal and brain circuitries involved in the processing of sensory afferents. The clear post-VIB effects found in the present study and others before [25, 26] are all suggestive signs of potential plastic adaptations within CNS networks involved in postural control. Future research should consider using neurophysiological tools (e.g. peripheral and central neurostimulation, electromyography, brain imagery) to further explore post-VIB effects and their underlying mechanisms.

## Variability of VIB-PR patterns and COP variables

One of the most intriguing discovery of our work is how individuals responded differently during this adaptative process of sensory re-weighting. Interestingly, previous studies showed increasing error estimates around their mean data when looking at measures obtained during the later phase compared to the early one [2, 22], in line with our findings. The duration of post-VIB effects also seem to vary greatly between persons in the literature, from a few seconds to up to an hour [15, 25, 26]. Neurophysiological variability is a hallmark of sensorimotor control [27–29], and typically illustrates how the nervous system benefits from a complex network of redundant and parallel pathways to produce a myriad of different movements and postures in accordance with our ever-changing goals and environment [30, 31]. The interpretation of VIB-PR variability has so far been restricted to a metrological standpoint [2], hence overlooking the core importance of physiological variability for motor control and learning [27–29]. Since the present work focused on methodological considerations and did not include neurophysiological measures to uncover these mechanisms, we cannot propose reliable explanations to account for such between-subject variability. Future studies are underway using electromyography and peripheral/central neurostimulators to address this question. Nevertheless, our study underscores the relevance of an individually tailored analysis of VIB-PR behaviors which adds complementary knowledge to group analysis.

Variability was further addressed in the present work using within- and between-subjects metrics focusing on COP measurements. Of note, this approach should not be confounded with SD observations from Fig 1, since it rather informs about the variability of global measures (maximal amplitude, mean velocity) collected through blocks of time (2s, 8s, 10s in the present study) instead of the precise evolution of raw COP positions in the anteroposterior plan. For example, Fig 2A clearly illustrates trial-to-trial variations of COP position in a single subject, but the highest min-max COP Amplitude reached during VIB (in this case when looking at the whole 10 s) did not vary much (6.20 cm for trial 1, 9.01 for T2, 6.07 for T3, 6.14 for T4 and 5.52 for T5). It is also interesting to note that despite Figs 1 and 2 clearly illustrating low individual and group SD for the early phase and high SD for the later phase, variability metrics of COP amplitude and velocity did not result in particularly clear contrasts. Variability was in some cases higher when looking at the *First 2 s* time-window, but most observations disappeared when correcting for the magnitude of the measure (i.e. expressing in % of the pooled mean). Three main conclusions are possible to draw from our results: (i) a certain level of within- and between-subjects variability exists for VIB-PR COP measurements, which is not particularly high (mostly below 30%) even when analyzing small time-windows; (ii) there is no added value of realizing five instead of three trials; (iii) other analytical methods (moving average, other COP-derived variables, etc.) should be explored for more adequately representing the contrasting variations observed in raw COP displacements between the early and later phase.

## Limitations

The interpretation and generalization of the study's findings are limited to healthy and young populations. The functional/physiological relevance of the proposed analytical methods will have to be investigated further using experimentally induced as well as pathological disruptions of sensorimotor functions. Also, participants were not instructed to try keeping / returning to their baseline position or actively fighting VIB-PR, but instead to simply keep balance. It is plausible that part of between-person variations of VIB-PR behaviors during the later phase can be ascribed to how each participant naturally reacted to VIB-PR. The impact of changing the directives given to participants should be investigated, for instance to test the influence of higher order cognitive processes on postural control and sensory re-weighting [32]. Future investigations could also use our analytical method based on early (first 2 sec) vs late responses (last 8 sec) to investigate the interplay between sensory and postural control systems with a combined used of different sensory disturbance methods (cutaneous VIB, vestibular stimulation, visual manipulations, etc.). The impact of using longer VIB durations, as well as post-VIB after-effects on the dynamic evolution of VIB-PR would also be of interest, as recently underscored [2]. Combining kinetic (force platform) with kinematic (3D motion) systems would be highly beneficial to understand further how the postural control system actually controls center of mass displacements through multi-joint coordination efforts [5]. This could also help identifying a potential source of between-subject variation of VIB-PR behaviors (e.g. different persons could use different ankle/knee/hip/trunk strategies to keep balance under VIB). Finally, in our study vibration of ankle tendons resulted in a transition between the early and later phases occurring 2 s after VIB start/stop. However, we highly recommend future work to first look at their mean group traces before selecting time-windows to analyze, since such transition would likely vary depending on the experiment (population, VIB location and parameters, postural task, etc.).

## Conclusion

To conclude, the present study encourages future research to consider the dynamic and variable nature of VIB-PR for COP analysis. By separating VIB-PR in distinct phases, the various underlying mechanisms could be more easily untangled, leading to new discoveries in the field of sensory re-weighting and postural control. Also, the identification of different VIB-PR behaviors could provide new insights on the variability in individual responses to a sensory disturbance. Finally, collecting five versus three trials per condition is more time-consuming and added no benefit in terms of variability of COP measures. Nevertheless, within- and between-individual variations of VIB-PR should be explored further. This is critically needed for getting a better grasp on the reliability/validity/responsiveness of our outcome measures. Variability could potentially be used as a proxy of how the postural control system adapts to changing contexts at the levels of the individual, the task or the environment.

## Acknowledgments

The authors would like to thank all the participants who took part in our study.

## Author Contributions

**Conceptualization:** Mohamed Abdelhafid Kadri, Hakim Mecheri, Martin Lavallière, Rubens A. da Silva, Louis-David Beaulieu.

**Data curation:** Mohamed Abdelhafid Kadri, Hakim Mecheri, Hugo Massé-Alarie, Rubens A. da Silva, Louis-David Beaulieu.

**Formal analysis:** Mohamed Abdelhafid Kadri, Emilie Bouchard, Lydiane Lauzier, Hakim Mecheri, Martin Lavallière, Hugo Massé-Alarie, Rubens A. da Silva, Louis-David Beaulieu.

**Investigation:** Mohamed Abdelhafid Kadri.

**Methodology:** Mohamed Abdelhafid Kadri, Rubens A. da Silva.

**Project administration:** Mohamed Abdelhafid Kadri, Louis-David Beaulieu.

**Supervision:** Mohamed Abdelhafid Kadri, Louis-David Beaulieu.

**Writing – original draft:** Mohamed Abdelhafid Kadri, Louis-David Beaulieu.

**Writing – review & editing:** Mohamed Abdelhafid Kadri, Emilie Bouchard, Lydiane Lauzier, Hakim Mecheri, William Bégin, Martin Lavallière, Hugo Massé-Alarie, Rubens A. da Silva, Louis-David Beaulieu.

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
