## [Decision Letter · Decision Letter 0]

9 Nov 2022

PONE-D-22-15288Distinctive phases and intra/inter-individual variability of vibration-induced postural reactions highlighted by spatiotemporal analyses of center of pressure displacementsPLOS ONE

Dear Dr. Beaulieu,

Thank you for submitting your manuscript to PLOS ONE. After careful consideration, we feel that it has merit but does not fully meet PLOS ONE’s publication criteria as it currently stands. Therefore, we invite you to submit a revised version of the manuscript that addresses the points raised during the review process.

I would like to apologise for the length of time it has taken to secure reviewers - it has been quite challenging. Given that this has taken this time, we will go ahead with the review submitted by this one reviewer. Can you please make sure that you address the reviewer's comments and in particular provide a clear explanation and justification for the protocol used to simulate postural responses to an increased risk of falling? Can you also please re-write the results section and ensure that it reports findings from the study. 

We look forward to receiving your revised manuscript.

Kind regards,

Aliah Faisal Shaheen

Academic Editor

PLOS ONE

2. Please ensure you specify in the Methods section of your manuscript text the type of informed consent provided by the participants (stated to be 'written' in the Ethics Statement of the online submission form).

Please also note that PLOS ONE has specific guidelines on code sharing for submissions in which author-generated code underpins the findings in the manuscript. In these cases, all author-generated code must be made available without restrictions upon publication of the work. Please review our guidelines at https://journals.plos.org/plosone/s/materials-and-software-sharing#loc-sharing-code and ensure that your code is shared in a way that follows best practice and facilitates reproducibility and reuse.

5. Please include your tables as part of your main manuscript and remove the individual files. Please note that supplementary tables (should remain/ be uploaded) as separate ""supporting information"" files.

Additional Editor Comments:

I would like to apologise for the length of time it has taken to secure reviewers - it has been quite challenging. Given that this has taken this time, we will go ahead with the review submitted by this one reviewer. Can you please make sure that you address the reviewer's comments and in particular provide a clear explanation and justification for the protocol used to simulate postural responses to an increased risk of falling? Can you also please re-write the results section and ensure that it reports findings from the study.

Reviewers' comments:

Reviewer's Responses to Questions

**Comments to the Author**

1. Is the manuscript technically sound, and do the data support the conclusions?

Reviewer #1: Yes

2. Has the statistical analysis been performed appropriately and rigorously? 

Reviewer #1: Yes

3. Have the authors made all data underlying the findings in their manuscript fully available?

Reviewer #1: No

4. Is the manuscript presented in an intelligible fashion and written in standard English?

Reviewer #1: Yes

5. Review Comments to the Author

Reviewer #1: The manuscript explores the analysis of postural responses following tendon vibrations, where responses are separated into distinct phases. As hypothesised, stronger postural responses are present during the early phase of vibration compared to the later ‘stabilisation’ phase, supporting the need for analysis to be separated into these distinct phases, as opposed to ‘whole-trial’ analyses. Overall, the justification for the research and methodological approach to answer the specific research question seem appropriate. However, there are a number of major/minor comments that remain.

Please see my detailed comments in the attached pdf.

6. PLOS authors have the option to publish the peer review history of their article (what does this mean?). If published, this will include your full peer review and any attached files.

Reviewer #1: No

---

## [Author Response · Author response to Decision Letter 0]

12 Dec 2022

All comments from the Reviewer have been responded in the file " Response to Reviewers".

---

## [Decision Letter · Decision Letter 1]

10 Jan 2023

Distinctive phases and variability of vibration-induced postural reactions highlighted by center of pressure analysis

PONE-D-22-15288R1

Dear Dr. Beaulieu,

We’re pleased to inform you that your manuscript has been judged scientifically suitable for publication and will be formally accepted for publication once it meets all outstanding technical requirements.

Kind regards,

Aliah Faisal Shaheen

Academic Editor

PLOS ONE

Additional Editor Comments (optional):

Reviewers' comments:

Reviewer's Responses to Questions

**Comments to the Author**

1. If the authors have adequately addressed your comments raised in a previous round of review and you feel that this manuscript is now acceptable for publication, you may indicate that here to bypass the “Comments to the Author” section, enter your conflict of interest statement in the “Confidential to Editor” section, and submit your "Accept" recommendation.

Reviewer #1: All comments have been addressed

2. Is the manuscript technically sound, and do the data support the conclusions?

Reviewer #1: (No Response)

3. Has the statistical analysis been performed appropriately and rigorously? 

Reviewer #1: (No Response)

4. Have the authors made all data underlying the findings in their manuscript fully available?

Reviewer #1: (No Response)

5. Is the manuscript presented in an intelligible fashion and written in standard English?

Reviewer #1: (No Response)

6. Review Comments to the Author

Reviewer #1: I would like to thank the authors for their detailed response to my initial review and for the changes made to the manuscript. I have no further comments and am happy to endorse this manuscript for publication.

7. PLOS authors have the option to publish the peer review history of their article (what does this mean?). If published, this will include your full peer review and any attached files.

Reviewer #1: No

---

## [Editor Report · Acceptance letter]

13 Jan 2023

PONE-D-22-15288R1 

Distinctive phases and variability of vibration-induced postural reactions highlighted by center of pressure analysis 

Dear Dr. Beaulieu:

I'm pleased to inform you that your manuscript has been deemed suitable for publication in PLOS ONE. Congratulations! Your manuscript is now with our production department. 

Kind regards, 

on behalf of

Dr. Aliah Faisal Shaheen 

Academic Editor

PLOS ONE